# Broadband near-infrared emission in silicon waveguides

Marcel W. Pruessner ®[1,3] ✉, Nathan F. Tyndall ®[1,3], Jacob B. Khurgin ®[2], William S. Rabinovich[1], Peter G. Goetz ®[1] & Todd H. Stievater ®[1,3]

Silicon photonic integrated circuit foundries enable wafer-level fabrication of entire electro-optic systems-on-a-chip for applications ranging from data-communication to lidar to chemical sensing. However, silicon's indirect bandgap has so far prevented its use as an on-chip optical source for these systems. Here, we describe a fullyintegrated broadband silicon waveguide light source fabricated in a state-of-the-art 300-mm foundry. A reverse-biased p-i-n diode in a silicon waveguide emits broadband near-infrared optical radiation directly into the waveguide mode, resulting in nanowatts of guided optical power from a few milliamps of electrical current. We develop a one-dimensional Planck radiation model for intraband emission from hot carriers to theoretically describe the emission. The brightness of this radiation is demonstrated by using it for broadband characterization of photonic components including Mach-Zehnder interferometers and lattice filters, and for waveguide infrared absorption spectroscopy of liquid-phase analytes. This broadband silicon-based source can be directly integrated with waveguides and photodetectors with no change to existing foundry processes and is expected to find immediate application in optical systems-on-a-chip for metrology, spectroscopy, and sensing.

Silicon photonics combines the technological maturity of complementary metal-oxide semiconductor (CMOS) microelectronics fabrication with the ultra-high bandwidth of light in nanophotonic waveguides. Microelectronics foundries can monolithically provide many of the key components for optical processing including high-speed electro-optic modulators, photodetectors, and low-loss waveguides. However, the materials commonly available in foundries (i.e., silicon, silicon dioxide, silicon nitride) are unable to emit or amplify optical radiation - a function critical for fully integrated optical systems[1]. Though significant progress in bringing light emission closer to the waveguides using techniques such as wafer-bonding[2], III–V semiconductor growth on silicon[3], or 2.5D packaging[4] has been made, these approaches—while enabling high output powers[2,3]—generally require substantial changes to the foundry process thereby complicating their integration with existing photonic components. Super-continuum sources have also been demonstrated in foundry-

compatible silicon nitride[5] and silicon-on-insulator (SOI) waveguides[6], but these sources require a high-power (off-chip) pump laser. While continued efforts to merge optical sources with silicon photonics have shown promise, the lack of a fully integrated, on-chip light source in silicon photonics continues to impede its maturation.

Despite silicon's indirect bandgap, optical emission from injected carriers has been observed previously in surface-normal geometries. Both electroluminescence from interband recombination in forward-biased diodes[7–10] and impact-ionization-based emission in reverse-biased p-(i)-n diodes[7,11–13] have been reported. However, the low quantum efficiency of the electrical-to-optical conversion has limited the performance of integrated electrically-pumped emitters for applications in digital communication. CMOS-compatible silicon light-emitting diodes (LEDs) have been explored previously[14,15], but they were either not integrated into a waveguide[16] or were narrowband and at wavelengths not used by data- or tele-communications systems[17].

[1]Naval Research Laboratory, Washington, DC, USA. [2]Department of Electrical and Computer Engineering, Johns Hopkins University, Baltimore, MD, USA. [3]These authors contributed equally: Marcel W. Pruessner, Nathan F. Tyndall, Todd H. Stievater. ✉e-mail: marcel.w.pruessner.civ@us.navy.mil

More recently, silicon photonic integrated circuits (PICs) have found potential applications beyond data-/tele-communications, including LiDAR, quantum computing, and chemical and biological sensing. Though some photonic sensing techniques, such as refractive-index sensing[18,19] or waveguide-enhanced Raman spectroscopy (WERS)[20,21] only require a fixed laser source, infrared (IR) spectroscopy requires a source with a broad optical bandwidth—often hundreds of nanometers. IR spectroscopies adapted for PICs include Fourier transform infrared spectroscopy (FTIR)[22], waveguide infrared absorption spectroscopy (WIRAS)[23,24], microring absorption spectroscopy[25,26], and tunable laser spectroscopy[27]. All of these demonstrations used an off-chip optical source, severely compromising the promise of integrated photonics.

Here, we demonstrate broadband emission from current injection within integrated silicon waveguides. We derive a theoretical framework based on the conventional Planck's radiation law but modified for hot carriers in a one-dimensional photonic (i.e., waveguide) geometry. We establish the utility of this source for the characterization of broadband waveguide components such as couplers and filters, as well as absorption spectroscopy of liquid analytes. This silicon waveguide emitter is fabricated in a standard process in a state-of-the-art silicon photonics foundry (AIM Photonics) enabling immediate integration into more sophisticated photonic integrated circuits.

## Results

### P-i-N waveguide emitter

The broadband emitter is integrated into a sub-micron rib waveguide, as shown in Fig. 1 and described in "Methods section" below. The silicon (Si) layer can be fabricated either as a ridge (fully etched) or a rib (half etched) waveguide designed for single-mode operation in the C-band. As shown in Fig. 1a, the undoped rib waveguide is placed laterally between a p-doped Si slab and an n-doped Si slab (Fig. 1b) to form the p-i-n emitter. Our design reconciles the need for the injection of high-velocity carriers through an undoped region that efficiently overlaps the optical waveguide mode. This requires an intrinsic region width and adjacent doping profile that minimizes dopant-based loss

and breakdown potential while maximizing carrier energy at the waveguide mode.

We measured the room temperature I–V characteristics of the p-i-n waveguides with 400 nm and 800 nm wide intrinsic regions as shown in Fig. 1c. The forward bias turn-on voltage is approximately 0.7 V and the reverse-bias breakdown voltages are −16 V (400 nm) and −28 V (800 nm). Surface-normal IR imaging of a p-i-n rib waveguide shows that the devices emit light for both forward and reverse bias. This light also couples to the modes of the rib waveguide, propagates down the ridge waveguide, and is then emitted surface-normal from a grating; or from a cleaved waveguide end-facet. The forward-bias emission from the grating appears to be emitted from a narrow spot, indicating a relatively narrow-band emission spectrum (Fig. 1d). When applying reverse-bias, however, we observe a broadened grating emission pattern (Fig. 1e). The angle of the grating emission is largely wavelength dependent (Supplementary Information, 4 Grating Emission), so a broad emission pattern suggests that many wavelengths are being coupled out of the grating. The surface-normal emission profile at the emitter is highly concentrated to the narrowest intrinsic regions for reverse bias (Fig. 1e) unlike the emission in forward bias (Fig. 1d) that is distributed through the i-region. As we will discuss, the IR images suggest distinct physical mechanisms for the emission depending on the bias point.

### Emission spectrum

We performed quantitative measurements of the emission spectrum using the setup depicted in Fig. 2a. The measured power spectra from the edge facet confirm narrowband emission near silicon's bandedge (1.1 eV) for forward bias (Fig. 2b) and broadband emission for reverse bias (Fig. 2c), in a device with a 400 nm wide intrinsic region. We note that both sets of spectra are limited by absorption in the silicon waveguide (wavelengths less than 1100 nm) between the emitter and the edge facet and by our spectrometer (detection range between 900 and 1590 nm). The reverse-bias emission is very stable over time (Supplementary Information, 11 Long-Term Emission) and likely extends far beyond 1590 nm in wavelength.

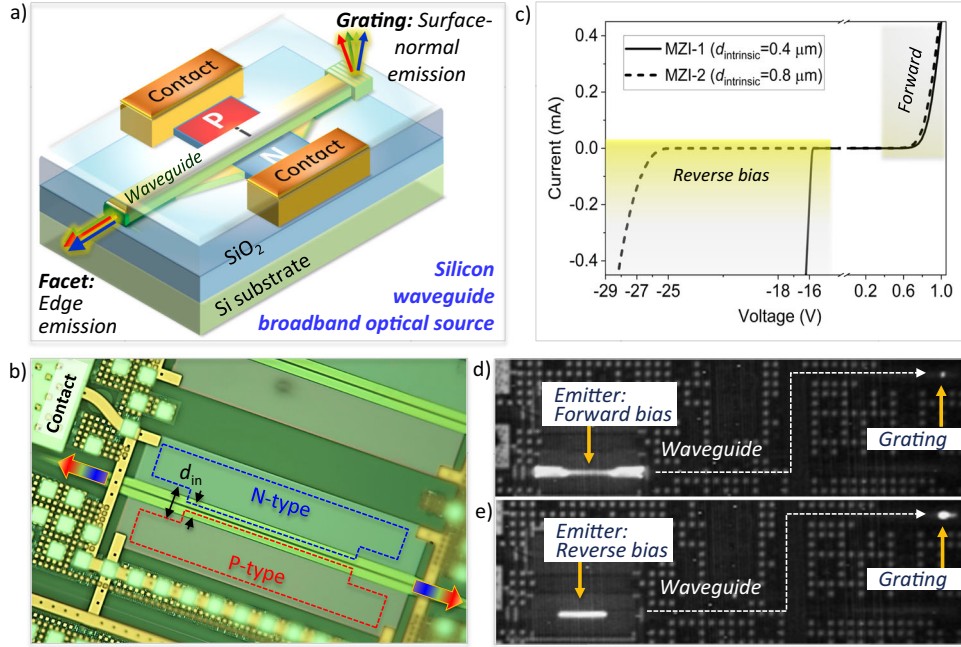

**Fig. 1 | P-i-N waveguide emitter. a** Device schematic, **b** fabricated p-i-n waveguide emitter device, **c** measured I–V curves for p-i-n waveguides with intrinsic widths of $d_{in}$ = 400 nm and 800 nm, **d** narrowband infrared (IR) grating emission from a

$d_{in}$ = 800 nm intrinsic region p-i-n waveguide under forward-bias, **e** broadband IR grating emission under reverse-bias. The optical microscope image in (**b**) has been adjusted for color temperature.

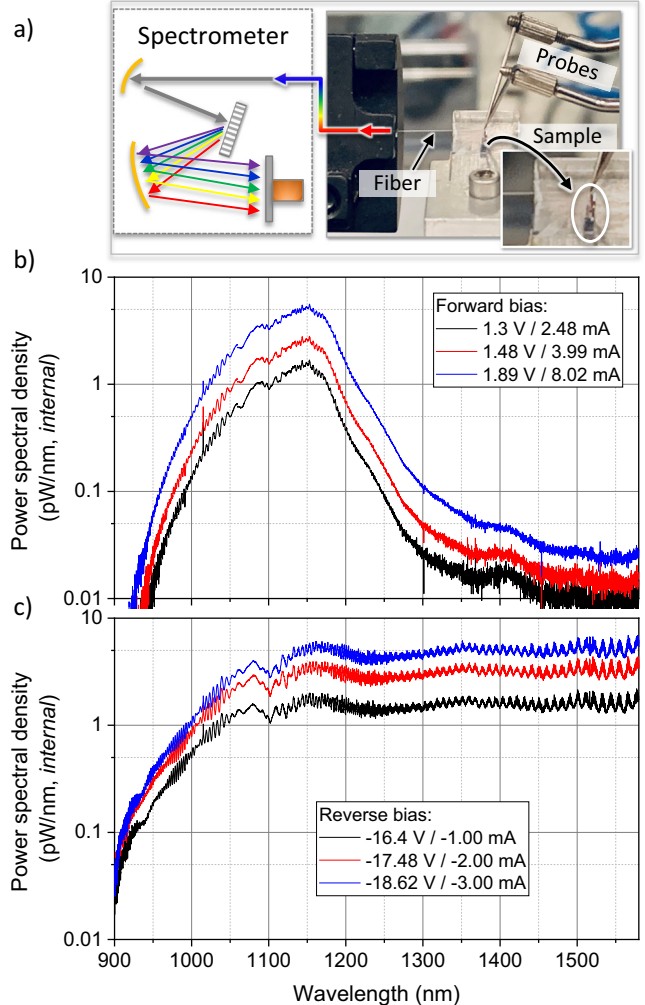

b)

Power spectral density (pW/nm, internal)

**Forward bias:**
— 1.3 V / 2.48 mA
— 1.48 V / 3.99 mA
— 1.89 V / 8.02 mA

c)

Power spectral density (pW/nm, internal)

**Reverse bias:**
— -16.4 V / -1.00 mA
— -17.48 V / -2.00 mA
— -18.62 V / -3.00 mA

Wavelength (nm)

**Fig. 2 | Measured emission spectrum. a** Measurement setup: electrical probes supply current to the device and a lensed fiber collects the emitted light that is sent to a spectrometer, **b** calibrated measured emission under forward bias, **c** emission under reverse bias.

## Emission mechanism

Surface-normal emission of light from biased p-(i)-n junctions in silicon has been previously observed[7,11,12,28,29]. Silicon's indirect bandgap prevents direct electron-hole recombination at the band edge, but a number of other mechanisms nevertheless allow for optical emission by injected carriers, albeit at lower quantum efficiency. For forward-bias, injected carriers in the depletion region can recombine with the aid of phonons (Supplementary Information, 7 Forward-Bias Emission) to produce emission that peaks near the bandedge[1,7,10,15,16] (1.1 eV). In a waveguide, this emission is then filtered by absorption in the silicon between the emission region and the collection fiber. As shown in the forward-bias measurements in Fig. 2b, this results in a peak waveguide emission that is slightly shifted to the red of the silicon bandedge. Forward-bias emission should then occur wherever injected current passes through the intrinsic region, which is exactly what is seen in the surface-normal hourglass emission profile in Fig. 1d.

For the reverse-bias emission spectra shown in Fig. 2c, the larger field in the intrinsic region accelerates injected electrons and holes to energies sufficient for impact ionization. Unlike the forward-bias spectra, these extreme-field (> 400 kV/cm) spectra show broadband emission spanning the wavelength range of our detector. Such low energy emission cannot be explained by interband recombination but instead must arise from intraband processes. We expect this intraband

emission to extend to the visible (to energies as high as the ionization energy, ≅2.3 eV)[7,12,28] but, as with the forward-bias spectra, at energies above the silicon bandedge any emission is absorbed by the waveguide.

Previous investigations of hot-carrier emission in surface-normal p-(i)-n diodes in silicon, combined with the frequency dependence of the measured power spectral density, suggests that the reverse-bias emission in our waveguides results from indirect intraband processes (Supplementary Information, 8 Reverse-Bias Emission). Indirect intraband photon emission could arise from either holes, electrons, or a combination of both. It requires carrier scattering from phonons or material defects to conserve momentum, as shown schematically in Fig. 3a.

The hot carriers are typically assumed to be in a non-equilibrium state with the lattice[12,28,30]. To estimate the emitted power spectral density in the waveguide mode, we assume that the hot-carrier plasma must be in thermal equilibrium with the modes of the electromagnetic field[31]. The Planck distribution combined with a one-dimensional (waveguide) density of states is the resulting power spectral density for a single mode and propagation direction[32]:

$$P(\nu) = \frac{h\nu}{e^{\frac{h\nu}{kT_{e,h}}} - 1} \tag{1}$$

where $\nu$ is the frequency of the optical radiation and $T_{e,h}$ is the hot carrier temperature (for electrons or holes). Note that the frequency dependence of Eq. (1) only depends on $\nu$ in the numerator, unlike the conventional (three-dimensional) Planck radiation law, which depends on $\nu^3$[32]. We can estimate the emissivity of the hot carrier plasma from its absorptivity, using Kirchoff's law, and use the well-known free-carrier absorption relationships to find the absorptivity[33]:

$$\alpha_{e,h}(\nu) = \frac{K_{e,h}N_{e,h}c^2}{\nu^2} \tag{2}$$

where $K_e = 1.0 \times 10^{-10}$ for electrons and $K_h = 2.7 \times 10^{-10}$ for holes, and $N_{e,h}$ is the carrier concentration. The carrier concentration can be approximated with the diode current and geometry: $N_{e,h} = I/(q\upsilon_{sat}L_{em}t_{em})$, where $I$ is the reverse current, $\upsilon_{sat}$ is the saturation velocity of the electrons or holes ($\upsilon_{sat} \sim 1 \times 10^5$ m/s in silicon[34]), and $L_{em}$ and $t_{em}$ are the length and thickness of the emitter's intrinsic region. Assuming an emissivity $\epsilon(\nu)$ equal to $\alpha(\nu)L_{em}\Gamma(\nu)n_g(\nu)/n_g^{Si}(\nu)$, where $\Gamma$ is the modal confinement factor within the intrinsic (emitter) region, $n_g$ is the waveguide mode group index, and $n_g^{Si}$ is the group index of silicon (Supplementary Information, 3 Rib Waveguide Properties), this model gives an emissivity-corrected power spectral density of

$$P(\nu)\epsilon(\nu) = \frac{Ihc^2}{q\upsilon_{sat}t_{em}\nu}\frac{\Gamma(\nu)n_g(\nu)}{n_g^{Si}(\nu)}\left(\frac{K_e}{(e^{\frac{h\nu}{kT_e}}-1)} + \frac{K_h}{(e^{\frac{h\nu}{kT_h}}-1)}\right) \tag{3}$$

assuming an equal density of electrons and holes. Note that this discussion applies to only a single electromagnetic mode of the waveguide, but both the TE$_{00}$ and TE$_{10}$ modes are supported by the rib below 1450 nm, as well as the TM$_{00}$ at wavelengths below 1080 nm (Supplementary Information, 3 Rib Waveguide Properties). Since the even Gaussian mode of our collection fiber mostly filters out the odd waveguide modes, we believe that the TE$_{10}$ mode is not significant in our measured spectra and is thus ignored in the analysis. The TM$_{00}$ mode is collected, but at attenuated levels due to silicon bandedge absorption.

This hot-carrier-emission (HCE) model has only two free parameters, the hot-hole and hot-electron temperatures. The measured power spectral density in Fig. 3b agrees well with the predicted power-spectral density from this model using a value of 9000 K for the carrier temperature, as shown in Fig. 3b. We emphasize that the lattice is near

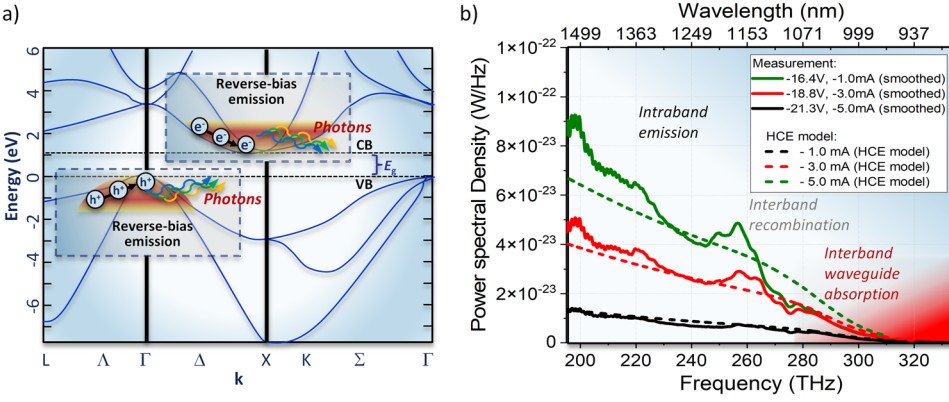

**Fig. 3 | Reverse-bias emission process. a** For reverse-bias, hot electrons (holes) in the silicon conduction (valence) band emit infrared photons via an intraband blackbody process. **b** Measured reverse-bias waveguide power spectral density vs. optical frequency. Also plotted is Eq. (3) for a carrier temperature of 9000 K, scaled by the silicon waveguide absorption between the emitter and the facet. The red shading in (**b**) indicates increased interband absorption by the output waveguide at wavelengths below silicon's bandedge.

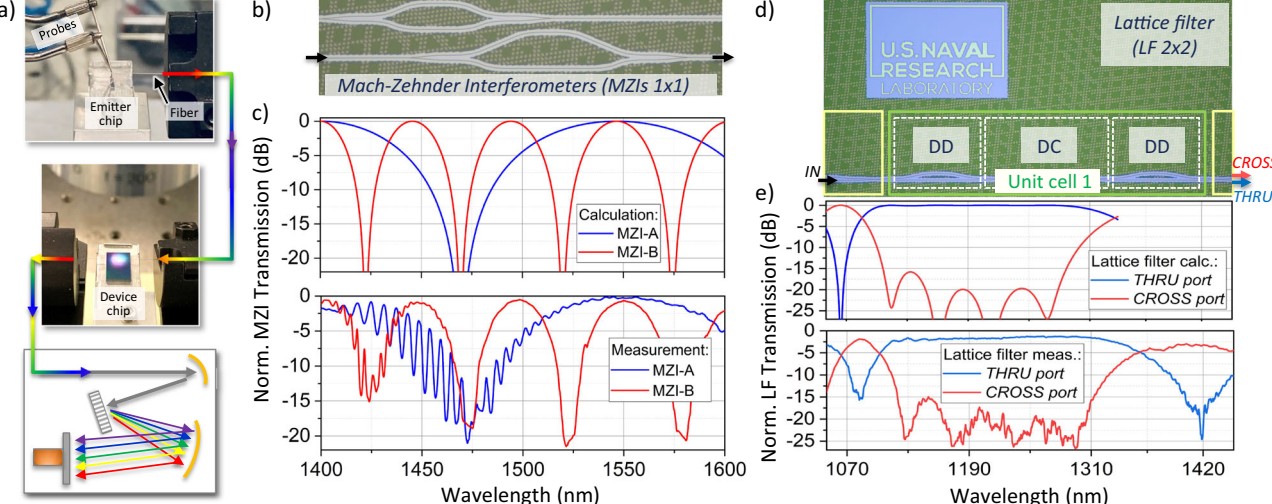

**Fig. 4 | Photonic device characterization using a broadband p-i-n emitter source. a** Measurement setup, (**b**) unbalanced 1 × 1 Mach–Zehnder Interferometers (MZIs) with different path length imbalances, (**c**) predicted and measured MZI spectra using broadband on-chip light source, (**d**) 2 × 2 lattice filter showing detail of unit cell comprising two differential delays (DD) and one broadband directional coupler (DC), (**e**) predicted and measured lattice filter response.

room temperature, while the hot carriers are highly-energetic. The model HCE spectrum also includes the waveguide absorption near the band edge to account for propagation losses between the emitter and the waveguide facet. It is likely that the actual hole and electron temperatures differ somewhat from each other due to variations in effective mass and saturation velocity. Though 9000 K is warmer than is typically surmised for hot carriers in silicon, it corresponds to a carrier energy of only 0.78 eV, consistent with Monte Carlo models of carriers in high fields[34]. Furthermore, this temperature is less than 20% higher than that reported previously in analyses of measured surface-normal emission from silicon diodes[12]. A carrier energy of 0.78 eV is also approaching the bandgap of silicon, which is a necessary condition to begin reverse breakdown of the diode. Our estimated hot-carrier temperature is also sensitive to uncertainties in other estimated parameters. For example, if either the free-carrier absorption or waveguide-fiber calibration were in error by a factor of two, the extracted carrier temperature could be closer to 5000 K. Despite these uncertainties, the HCE model shows excellent agreement with both the spectral shape and absolute intensity of the emission (Fig. 3b). In addition, this model predicts similar powers from devices with a diode

intrinsic region width that varies between 400 nm and 800 nm, which is largely consistent with our measurements (Supplementary Information, 6 Intrinsic Region Width).

Intraband hot-carrier emission would be expected to arise only from regions of the diode with a field sufficient to accelerate carriers to their saturation velocities. Thus, unlike the forward-bias case, reverse-bias based intraband HCE should only occur in the depleted regions of the diode with the narrowest gap (highest field) between the anode and the cathode. Such a surface-normal emission profile is exactly what is observed in Fig. 1e.

## Application: integrated device characterization

The broadband hot-carrier emission in a sub-micron silicon waveguide is of order nanowatts and can be used for on-chip characterization of integrated photonic components without the need for an off-chip tunable laser or broadband source. To demonstrate this, we couple light from our silicon emitter to a polarization maintaining (PM) optical fiber that connects to a second PIC chip that is also fabricated at AIM Photonics using the same process flow and fabrication tools (Fig. 4a). This second PIC chip has various broadband integrated

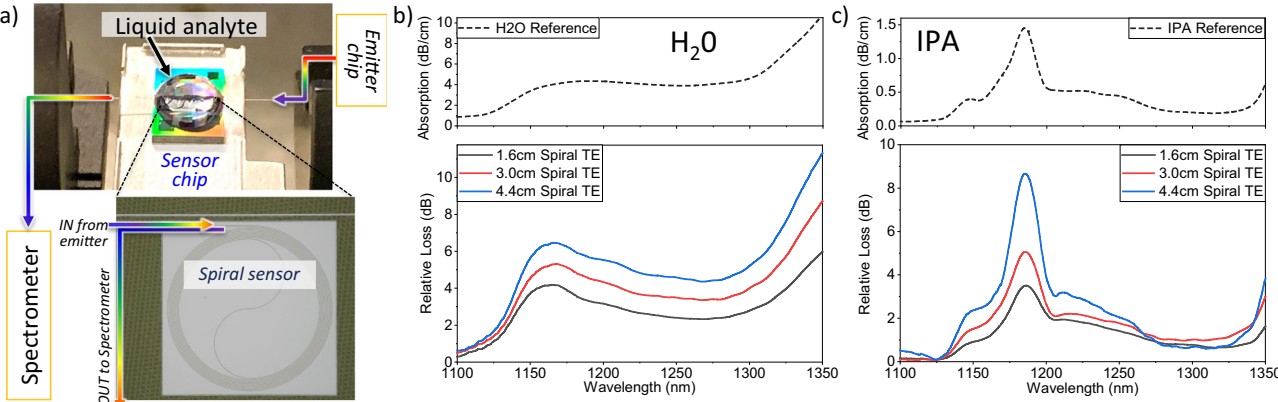

**Fig. 5 | Absorption spectroscopy of liquid analytes. a** Measurement setup showing the emitter chip, sensor chip with liquid analyte, detail of the spiral waveguide sensor, and the spectrometer; the individual components are connected via optical fiber, **b** reference spectrum[40] and measured WIRAS spectrum of water, **c** reference spectrum[41] and measured WIRAS spectrum of isopropyl alcohol (IPA).

photonic components, which can be fabricated in the same PIC as the silicon emitter. We characterize two optical devices: a pair of unbalanced Mach–Zehnder interferometers (MZIs)[35]; and a lattice filter[36] for passive filtering. The output from these PIC components is sent via PM fiber to the same spectrometer as described above.

The MZIs are single-input/output devices (1 × 1 MZIs) as shown in Fig. 4b designed for single-mode operation in the C/L bands. They comprise silicon nitride (SiN) waveguides that are 220 nm thick and 1500 nm wide. Input light splits via a waveguide Y-splitter into two separate, unbalanced waveguide paths. The two paths recombine, resulting in an interference spectrum determined by the path length imbalance of the two arms. Our measurements span a wavelength range from 1400 to 1600 nm, which is difficult to achieve in a single off-chip tunable laser system, and, until this work, not possible to achieve with a single *on-chip* source. The measurements in Fig. 4c show interference spectra from two devices. MZI-B has a path length imbalance exactly three times that of MZI-A, resulting in an increase in the interference order. The interference maxima and minima wavelengths enable the unambiguous determination of $n_{eff}$ for the TE$_{00}$ mode[35,37]. The spectra also exhibit a high-frequency oscillation at short wavelengths that is attributed to beating between the TE$_{00}$ and TE$_{10}$ modes, since the SiN waveguides support the TE$_{10}$ mode at $\lambda < 1450$ nm. Narrower waveguides would enable single-mode operation at shorter wavelengths, including for a Si emitter, to ensure emission into the fundamental waveguide mode at any wavelength band of interest. MZIs have become critical PIC components for in-situ material process characterization and statistical analysis for tracking wafer-scale variability of photonic geometries and indices[35,38].

The second photonic component we characterized is a four-port lattice filter[36]. This filter enables large passbands with narrow rejection bands and is useful for e.g. filtering out the laser pump signal in on-chip WERS[39] or fluorescence spectroscopy. The filter response is determined from cascaded unit cells, each of which consists of a differential delay (DD) and a broadband directional coupler (DC) section (Fig. 4d). This particular device was designed to reject (pass) a laser at 1064 nm and pass (reject) light at longer wavelengths in the thru (cross) port. The measured spectra span a wavelength range from 1050 to 1450 nm and confirm a large passband of 180 nm (in the thru port) with relatively narrow rejection bands centred at 1082 nm and 1420 nm (Fig. 4e).

**Application: liquid-phase analyte sensing**

Our waveguide emitter also enables waveguide infrared absorption spectroscopy (WIRAS) for chemical sensing[24]. Similar to the component characterization, we couple the silicon emitter via PM fiber to a sensing chip (Fig. 5a); future designs can incorporate the silicon emitter and sensing waveguide on a single substrate. WIRAS requires a single-mode, low-loss waveguide with a sensing trench for exposure of the waveguide optical mode to the liquid analyte. The trench region (the square in the spiral waveguide sensor in Fig. 5a) has the top SiO$_2$ cladding removed to enable access to the evanescent field of the propagating mode. The SiN waveguides used here are 220 nm thick and 800 nm wide and are patterned into spirals in the sensing trench. We collect the IR light transmitted through the sensing spiral with a second PM fiber and send it to the spectrometer.

Two liquid analytes served as proof of principle for this work: water (H$_2$O) and isopropyl alcohol (IPA). For each, we pipetted a drop onto the spiral waveguide as shown in Fig. 5a. We then measured the transmission through three different lengths of waveguide spirals whose transmission spectra is modified by the presence of the liquid analyte. The measured WIRAS spectra from H$_2$O and IPA are displayed in Fig. 5b,c, respectively, along with reference spectra[40,41]. The measured spectra are normalized and, as expected, the absorption increases with the spiral length. Our measured data agree very well with the reference spectra, enabling clear identification of both analytes. These measurements confirm that the light from the silicon waveguide emitter is sufficiently bright to perform WIRAS on common liquid analytes and demonstrates the practical utility of our waveguide-integrated silicon light source for chip-scale sensing applications.

## Discussion

The integrated power measured over the detection range of the spectrometer (up to $\lambda = 1590$ nm) is 5.1 nW at 5 mA reverse bias current, corresponding to a measured efficiency of approximately $1.0 \times 10^{-6}$ W/A and a quantum efficiency of $1.1 \times 10^{-6}$. The efficiency compares favorably with previous reports of emission from silicon devices, and the integrated optical power emitted vs. input electrical power is approximately twice as large as previous reports for forward-bias emission[17]. In addition, the integrated internal intensity of the emitter, 4100 mW/cm² at 5 mA, is an order of magnitude larger than the previously reported record value for a silicon emitter, 600 mW/cm²[42]. Emission peaks near 250–260 THz seen in Fig. 3b are consistent with *n*- and *p*-type dopants introducing new mid-gap energy levels[29], although other factors including multi-mode behavior at these wavelengths could also play a role.

Finite-element models of lattice heating due to diode resistance (Supplementary Information, 9 Joule Heating and Reliability) indicate that the local lattice temperature at the emitter rises by 250 K in our measurements, and thermal failure at the diode is likely to pose the ultimate limit to the achievable emitted power (Supplementary

Information, 9 Joule Heating and Reliability). Though the heating is localized to the biased diode, thermal cross-talk due to lateral heat flow to adjacent devices may impact scalability. Future thermal isolation strategies using trenches and undercuts[43] will mitigate this concern.

Since it is impossible to establish thermal equilibrium between hot carriers and a bulk device (lattice or otherwise) without cooling the carriers, reports of a specific hot carrier temperature are necessarily indirect. Electron and hole energies of 0.5 eV at fields of $10^5$/cm (at a 300 K lattice temperature) were reported in ref. 34. Extrapolated to our fields that exceed $3{\times}10^5$ V/cm (silicon's breakdown field), carrier energies of 0.8 eV are quite reasonable. Such energies correspond to hot carrier temperatures of 9000 K. This carrier energy is also closer to the silicon bandgap, and a carrier energy that exceeds the bandgap is required to initiate avalanche breakdown (1.6 eV is our estimate based on approximately 1.4x the silicon bandgap of 1.1 eV). We hypothesize that during avalanche breakdown there is a broad range of carrier temperatures (energies) due to the stochastic nature of the process. These carrier energies would range from as high as 1.6 eV down to fractions of an eV. The hotter carriers, which may not be the most likely component of this statistical distribution, would contribute disproportionately to the optical emission due to the temperature dependence ($T^4$) of Planck radiation.

The waveguide hot-carrier emission is limited at short wavelengths by silicon band edge absorption, and at long wavelengths by substrate leakage, bend loss, or material absorption. For the rib waveguides considered here, transmission out to $\lambda > 3.0\,\mu m$ should be possible. The reverse-bias emission from our silicon optical source is sufficient for high-fidelity photonic device characterization and mode cutoff determination. Even higher signal and signal-to-noise levels are possible by integrating the source and components on the same PIC, avoiding multiple fiber-to-waveguide coupling losses (Supplementary Information, 5 Post-Processing: Waveguide Facets).

Regarding absorption spectroscopy for chemical sensing (Fig. 5), the wavelength range of our broadband source exceeds that of off-chip tunable lasers and LEDs and thus offers unique capabilities for chip-scale sensors. Further, the signal-to-noise ratio (SNR) of our source is adequate for the detection of liquid analytes (Supplementary Information, 13 Source Signal-to-noise Ratio). Improvements to the limit of detection (LOD), potentially by several orders of magnitude can be obtained by, e.g., eliminating the fiber-chip coupling losses by incorporating both the emitter and spiral sensor on a common substrate, increasing the spiral waveguide length, and by operating the Si emitter at higher output powers. Although the spectroscopy demonstrations required an off-chip spectrometer, many efforts are underway to develop integrated spectrometers[22,44–46] in silicon and silicon nitride platforms. It is also important to note that compact, uncooled, or thermoelectrically-cooled spectrometers are now available with very similar performance as benchtop liquid-nitrogen cooled systems[47]. In fact, our measurements of the reverse-bias HCE spectra with a handheld spectrometer show nearly identical signal and signal-to-noise as that obtained in the benchtop system described above (Supplementary Information, 10 Spectrometer Comparison).

In conclusion, we experimentally demonstrated a broadband-IR silicon light source fabricated in a state-of-the-art 300-mm PIC foundry (AIM Photonics) and integrated into a nanophotonic waveguide. The emission is described using a developed theoretical framework that combines Planck radiation in one dimension with hot carriers in a reverse-biased semiconductor diode. Our IR measurements clarify the emission mechanism from reverse-biased p-i-n diodes as arising from indirect intraband carrier scattering. This framework suggests significantly more emitted power is yet possible from these devices provided that thermal runaway can be avoided. This is, to our knowledge, the first integrated waveguide broadband silicon emitter, and the highest reported intensity of any silicon emitter. Notably, since the fabrication has been carried out in a PIC foundry, the source can be immediately included in the component library for straightforward integration with other on-chip photonic devices.

As the applications for silicon photonics expand beyond data-/tele-communications, a broadband on-chip light source will become increasingly valuable. This source can be used for in-situ broadband component characterization[48], for wafer-scale process control[35,38], for IR absorption spectroscopy[24,27], or for Fourier-transform IR spectroscopy[49]. When combined with on-chip spectrometers, spectral filters, or even placed inside of waveguide cavities[17], this broadband optical source will fill a critical gap towards fully integrated spectroscopy systems on a chip.

## Methods
### Waveguide fabrication
The devices are fabricated at AIM Photonics using a similar process to that of the base active PIC multi-project wafer service. The silicon layer is 220 nm thick and can be fabricated either as ridge (full etch, 480 nm wide) or rib (half-etch, 550 nm wide) waveguides. The waveguides are clad in $SiO_2$. The ridge waveguides outside of the emission region adiabatically convert to rib waveguides in the $60\,\mu m$-long emission region. As shown in Fig. 2, the undoped rib waveguide is placed laterally between a p-doped Si slab and an n-doped Si slab. Both slabs are ohmically contacted by metal layers used for charge injection and biasing. The width of the undoped gap between the n- and p-doped regions ($d_{in}$) is 400 nm or 800 nm in the devices presented here.

### Emission spectrum measurement setup
The emission spectrum was measured using the setup depicted in Fig. 2a. First, broadband emission generated by the p-i-n waveguide couples to a lensed polarization-maintaining (PM) fiber (Oz Optics TPMJ) via a cleaved waveguide facet. The fiber slow axis is aligned parallel to the sample plane. Cleaved facets are preferable to grating couplers because the latter has a narrow spectral response ($FWHM = 30$ nm, Supplementary Information, 4 Grating Emission), whereas the former can efficiently couple light over hundreds of nanometers. The light propagates via PM fiber to a 0.5-m Czerny-Turner grating spectrometer (Princeton Instruments SP2558) with a liquid nitrogen-cooled InGaAs detector. We used the calculated fiber-facet coupling (Supplementary Information, 5 Post-Processing: Waveguide Facets), and the spectrometer grating and detector efficiency curves to estimate the absolute (internal) power in the waveguide.

## Data availability
Data sets generated during the current study may be available from the corresponding author upon request. Data release is subject to approval by the US Naval Research Laboratory.

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

## Acknowledgements

The authors acknowledge support from the the Office of Naval Research (NRL WU 1AB2) (M.W.P., N.F.T., W.S.R., P.G.G., and T.H.S).

## Author contributions

M.W.P., N.F.T., and T.H.S. proposed and conceived this project; M.W.P. designed the devices; N.F.T. performed the chip layout; M.W.P. and T.H.S. performed device post-processing; M.W.P., N.F.T., and T.H.S.

performed the measurements and data analysis; T.H.S. performed the theoretical analysis with assistance from J.B.K. and W.S.R.; P.G.G. provided guidance for foundry integration. M.W.P., N.F.T., and T.H.S. wrote the manuscript with input and comment from all co-authors.

## Competing interests

The authors declare no competing interests.
