## [Peer Review File · Nature Communications]

Broadband Near-Infrared Emission in Silicon WaveguidesEditorial Note: This manuscript has been previously reviewed at another journal that is not operating a transparent peer review scheme. This document only contains reviewer comments and rebuttal letters for versions considered at *Nature Communications*.

REVIEWER COMMENTS

Reviewer #2 (Remarks to the Author):

The authors present the design and characterization of an integrated silicon emitter based on a p-i-n junction. They demonstrate broadband light emission covering the near-infrared region including telecom bands. The integration of light sources on silicon photonic circuits is a key-point in the development of scalable photonic circuits. The authors explain the emission in reverse bias with a new theoretical model of intraband emission from hot carriers. Practical demonstrations of component characterizations and spectroscopy using this silicon source are presented, highlighting the potential of this approach. I believe the work presented in the manuscript is of great interest for the photonic community as many applications require low-cost, scalable broadband sources. The previous comments made by the reviewers raised important points but in my opinion the authors already answered most of them in a satisfactory way.

I recommend publication of this work with minor revisions.

My comments are the following:

1. The authors could include more details about the theoretical model in the discussion in order to avoid any confusion for the reader. I find the authors' response to reviewer 1 (Comment 2) in the rebuttal letter of december 2023 satisfying and part of the answer can be used in the main manuscript to stress the validity of the model and its interpretation.
2. The authors show a proof-of-concept of analyte sensing. The result is very interesting and already convincing but a short discussion about the limitations of their device for spectroscopy applications and the possible ways of improvement would strengthen the argument on the utility of the broadband emitter.

In conclusion, the authors present an interesting work towards the integration of broadband sources on silicon photonic integrated circuits. The relevance of the approach and the achievements demonstrated justify publication of their work.

Reviewer #3 (Remarks to the Author):

The authors have not properly addressed two of my main concerns, namely the applicability of the approach and the need for a systematic study to find the optimal geometry of the waveguide. There is no clear experiment showing that the approach could be of real interest. In addition, not being able to provide an experimental investigation of the optimal waveguide geometry reinforces the impracticality nature of the approach. As a result, I cannot recommend publishing this paper in a high impact journal.

AUTHORS' RESPONSE TO REVIEWER COMMENTS – *ROUND 3*

Reviewer #1 (Remarks to the Author):

No comments given.

Reviewer #2 (Remarks to the Author):

GENERAL COMMENT: The authors present the design and characterization of an integrated silicon emitter based on a p-i-n junction. They demonstrate broadband light emission covering the near-infrared region including telecom bands. The integration of light sources on silicon photonic circuits is a keypoint in the development of scalable photonic circuits. The authors explain the emission in reverse bias with a new theoretical model of intraband emission from hot carriers. Practical demonstrations of component characterizations and spectroscopy using this silicon source are presented, highlighting the potential of this approach. I believe the work presented in the manuscript is of great interest for the photonic community as many applications require lowcost, scalable broadband sources.

The previous comments made by the reviewers raised important points but in my opinion the authors already answered most of them in a satisfactory way.

I recommend publication of this work with minor revisions.

AUTHORS' RESPONSE: We thank *reviewer 2* for the careful review of our work, and also for the adjudication of our authors' response and revisions from *round 1* and *round 2* of review. We agree with *reviewer 2* that this work is “of great interest for the photonics community as many applications require lowcost, scalable broadband sources.” It is our hope that the broadband emitter presented in this work, if integrated in a foundry offering, can find many practical application areas ranging from on-chip device metrology to chemical sensing.

COMMENT 1: My comments are the following: The authors could include more details about the theoretical model in the discussion in order to avoid any confusion for the reader. I find the authors' response to reviewer 1 (Comment 2) in the rebuttal letter of december 2023 satisfying and part of the answer can be used in the main manuscript to stress the validity of the model and its interpretation.

AUTHORS' RESPONSE 1: We agree that some details from our rebuttal (18 December 2023) can help clarify the discussion of the emission and modeling. We have therefore added the following:

Page 7, last paragraph (“This hot-carrier emission (HCE)...”): added “**We emphasize that the lattice is near room temperature, while the hot carriers are highly-energetic.**”

Page 11, Added a new paragraph: “**Since it is impossible to establish thermal equilibrium between hot carriers and a bulk device (lattice or otherwise) without cooling the carriers, all claims of a specific hot carrier temperature are necessarily indirect. Electron and hole energies of 0.5 eV at fields of 10^5 V/cm (at a 300 K lattice temperature) were reported in [34]. Extrapolated to our fields that exceed 3×10^5 V/cm (silicon's breakdown field), carrier energies of 0.8 eV are quite reasonable. Such energies correspond to hot carrier temperatures of 9000 K. This carrier energy is also closer to the silicon bandgap, and a carrier energy that exceeds the bandgap is required to initiate avalanche breakdown (1.6 eV is our estimate based on approximately 1.4x the silicon bandgap of 1.1 eV). We hypothesize that during avalanche there is a broad range of carrier temperatures (energies) due to the stochastic nature of the process. These carrier energies would range from as high as 1.6 eV down to fractions of an eV. The hotter carriers, which may not be the most likely component of this statistical distribution, would contribute disproportionately to the optical emission, due to the temperature dependence (T^4) of Planck radiation.**”

COMMENT 2: The authors show a proof-of-concept of analyte sensing. The result is very interesting and already convincing but a short discussion about the limitations of their device for spectroscopy applications and the possible ways of improvement would strengthen the argument on the utility of the broadband emitter.

AUTHORS' RESPONSE 2: As mentioned in our response following *round 1* of review, the measurement-specific SNR is approximately 31 resulting in a limit of detection (LOD) for IPA of 3% – already low enough to be relevant for applications such as analyzing mixtures of unknowns and monitoring analyte concentrations in liquid samples. This LOD is determined by the signal-to-noise ratio (SNR) of the combined sensing *system* (i.e. the optical source from the first chip, fiber-coupling from the source chip to the sensor chip, and the second chip containing the spiral waveguide sensor used for absorption spectroscopy).

Based on the SNR discussion in the manuscript and in the supplementary information, a number of improvements can be made to increase the SNR and hence the LOD of the absorption spectroscopic sensor. First, the Si emitter source and the spiral waveguide sensor can be integrated on a single chip. This eliminates two fiber-chip coupling interfaces, each of which has a loss of >10 dB (see Fig. 10b in the Supplementary Information). The integration of the emitter and the sensing spiral on one substrate can thus potentially realize an improvement in the SNR and LOD by 25 dB.

Second, the longest spiral waveguide had a length of 4.4 cm. As can be seen from the absorption spectra of both DI H₂O and IPA, an increase in waveguide length results in an increase in the absorption length and a stronger absorption feature. Based on the measurements in Fig. 5 of the manuscript, the waveguide propagation loss is sufficiently low (down to 0.1 dB/cm, see <https://doi.org/10.1117/12.2582529>) so that increased LOD can be obtained by simply increasing the spiral waveguide sensor length. Of course, the spiral length cannot be increased indefinitely since waveguide propagation loss will eventually dominate.

Finally, the emitter power is currently limited by lattice heating (see section 9 Joule Heating and Reliability in the Supplementary Information). For e.g. the IPA absorption spectroscopy measurements in Fig. 5(b) the emitter was operated at a reverse-bias voltage of -17.49 V (-1.95 mA) – well below the thermal lattice limit for this device. In fact, we've operated the device at twice this electrical power (i.e. -19.40 V / -3.50 mA) without issue, suggesting that the measurements could have been done with twice the optical power and a corresponding 3 dB increase in LOD. Further device optimization to limit thermal runaway (e.g. via active cooling of the chip) may enable a further increase in optical power and LOD.

Taken together, the above three improvements in coupling loss, spiral sensor length, and emitter power suggest that the LOD can potentially be improved by several orders of magnitude, perhaps up to 30 dB.

Fig. A: Si Emitter operated at 35 mW reverse bias (*blue curve*) and 68 mW (*green curve*).

CHANGES TO MANUSCRIPT:

Page 11, third paragraph (“The waveguide hot-carrier...”): “Even higher signal and signal-to-noise levels are possible by integrating the source and components on the same PIC, avoiding multiple fiber-to-waveguide coupling losses (currently >10 dB per fiber-chip coupling, see Supplementary Information).”

Page 11, fourth paragraph (“Regarding absorption spectroscopy...”): We summarize the above improvements to the sensor LOD: “Improvements to the LOD, potentially several orders of magnitude, can be obtained by e.g. eliminating the fiber-chip coupling losses by incorporating both the emitter and spiral sensor on a common substrate, increasing the spiral waveguide length, and by operating the Si emitter at higher output powers.”

COMMENT 3: In conclusion, the authors present an interesting work towards the integration of broadband sources on silicon photonic integrated circuits. The relevance of the approach and the achievements demonstrated justify publication of their work.

AUTHORS’ RESPONSE 3: We thank *reviewer 2* again for the careful reading of our manuscript.

Reviewer #3 (Remarks to the Author):

COMMENT 1: The authors have not properly addressed two of my main concerns, namely the applicability of the approach [...]. There is no clear experiment showing that the approach could be of real interest.

AUTHORS’ RESPONSE 1: We feel that this comment was adequately addressed in the manuscript as well as in our response to *reviewer 3* in *round 1* and *round 2* of review. **This work conclusively demonstrates that our developed broadband Si emitter has sufficient power to enable practical applications** ranging from on-chip metrology of photonic integrated circuit (PIC) components (e.g. Mach-Zehnder interferometers or MZIs; and lattice filters) to chemical sensing (e.g. absorption spectroscopy of isopropyl alcohol or IPA). Concerning on-chip metrology, our characterization of MZIs and lattice filters showed that our broadband source is indeed sufficiently bright to accurately characterize the transmission spectrum of these devices. As for chemical sensing, our analysis showed that we are able to sense IPA (liquid phase analyte) down to concentrations of 3%. Our response to *Comment 2* from *Reviewer 2* discusses several methods by which the limit of detection (LOD) can be improved by potentially several orders of magnitude. Therefore, we feel that the two demonstrations presented in this work conclusively show that our broadband source and our waveguide-integrated p-i-n emitter approach enable practical applications. Furthermore, there is substantial interest in this work; e.g. *reviewer 2* states “The authors show a proof-of-concept of analyte sensing. The result is very interesting and already convincing.” In addition, this work was elevated to an invited presentation at CLEO in 2023 attesting to the high level of excitement and interest in this work by the active optical sensing community.

COMMENT 2: [...] and the need for a systematic study to find the optimal geometry of the waveguide. [...] In addition, not being able to provide an experimental investigation of the optimal waveguide geometry reinforces the impracticality nature of the approach. As a result, I cannot recommend publishing this paper in a high impact journal.

AUTHORS' RESPONSE 2: We disagree with *reviewer 3's* comment stating that a “systematic study to find the optimal geometry of the waveguide” is required for proving the practicality of our broadband integrated silicon waveguide emitter. **An investigation of the optimal waveguide geometry, while potentially a useful future study, does not change the fundamental results of this work,** namely: 1) the foundry-compatible design of an all-silicon waveguide-integrated broadband optical source, 2) the experimental characterization and theoretical modeling of this source, and 3) the practical application of this source to PIC metrology and chemical sensing. Furthermore, the study of the optimal geometry is an open-ended problem statement since the optimal geometry will depend on the wavelength of operation and intended application. We already have addressed the waveguide properties *in detail* in the Supplementary Information section **3 Rib Waveguide Properties** and the fiber-waveguide coupling in section **5 Post-Processing: Waveguide Facets**. In particular, our measurements and simulations show that the Si emitter waveguide is dominated by the TE-modes since the TM₀₀-mode is cut-off at wavelengths $\lambda > 1100$ nm. As mentioned in the manuscript on p. 11, “The waveguide hot-carrier emission is limited at short wavelengths by silicon band edge absorption, and at long wavelengths by substrate leakage, bend loss, or material absorption.” The waveguides can be designed to enable low-loss transmission at longer wavelengths, but this generally results in multi-mode waveguides at shorter wavelengths. Single-mode waveguides at short wavelengths, however, result in poorly-confined modes at longer wavelengths resulting in substrate leakage and large propagation loss. Therefore, the “optimal geometry” is dependent on the specific wavelength band and application.

Further, our characterization of photonic components (unbalanced MZIs and lattice filters) as well as liquid chemical analyte sensing using absorption spectroscopy (DI water and IPA) clearly demonstrate the practical application of our Si emitter as an on-chip broadband optical source. We therefore feel that the fabrication of new Si emitter devices and their characterization to determine an optimal geometry falls outside the scope of this paper, and an experimental optimization study of the waveguide geometry can be the focus of future work. Regarding the *reviewer's* assertion of the “impracticality nature” of our approach, we feel that fabrication in a CMOS-compatible foundry, characterization of multiple chips and devices (see “Table: Summary of chips and devices measured in this work” in our authors' response from 6 November 2023), and **two demonstrations of using our emitter to address distinct problems of interest to the integrated photonics community** (photonic component characterization in Fig. 4 and chemical sensing via absorption spectroscopy in Fig. 5) **have conclusively demonstrated the practicality of this work.**

REVIEWERS' COMMENTS

Reviewer #2 (Remarks to the Author):

The authors answered all my comments and I support publication of the article in *Nature Communications*.

AUTHORS' RESPONSE TO REVIEWER COMMENTS – *ROUND 4*

Reviewer #1 (Remarks to the Author):

No comments given.

Reviewer #2 (Remarks to the Author):

GENERAL COMMENT: Reviewer #2 (Remarks to the Author):

The authors answered all my comments and I support publication of the article in Nature Communications.

AUTHORS' RESPONSE: We again thank *reviewer 2* for the additional insightful comments and suggestions during *round 3* of review that have helped to improve this manuscript.

Reviewer #3 (Remarks to the Author):

No comments given.

AUTHORS' RESPONSE: We appreciate *reviewer 3's* comments and suggestions over multiple rounds of review, even if we did not agree on all points that were raised.